

# Connecting laboratory behavior to field function through stable isotope analysis

Mael G. Glon[1], Eric R. Larson[2] and Kevin L. Pangle[1]

[1] Department of Biology, Central Michigan University, Mt Pleasant, MI, United States
[2] Department of Natural Resources and Environmental Sciences, University of Illinois at Urbana-Champaign, Urbana, IL, United States

## ABSTRACT

Inherent difficulties of tracking and observing organisms in the field often leave researchers with no choice but to conduct behavioral experiments under laboratory settings. However, results of laboratory experiments do not always translate accurately to natural conditions. A fundamental challenge in ecology is therefore to scale up from small area and short-duration laboratory experiments to large areas and long durations over which ecological processes generally operate. In this study, we propose that stable isotope analysis may be a tool that can link laboratory behavioral observations to past field interactions or function of individual organisms. We conducted laboratory behavioral assays to measure dominance of invasive rusty crayfish, *Orconectes rusticus*, and used stable isotope analysis to hindcast trophic positions of these crayfish under preceding natural conditions. We hypothesized that more dominant crayfish in our assays would have higher trophic positions if dominance were related to competitive ability or willingness to pursue high-risk, high-reward prey. We did not find a relationship between crayfish dominance and trophic position, and therefore infer that laboratory dominance of crayfish may not necessarily relate to their ecology in the field. However, this is to our knowledge the first attempt to directly relate laboratory behavior to field performance via stable isotope analysis. We encourage future studies to continue to explore a possible link between laboratory and field behavior via stable isotope analysis, and propose several avenues to do so.

Corresponding author
Mael G. Glon, glon1mg@cmich.edu

# INTRODUCTION

Animal behavior is inherently linked with the fields of ecology and evolution (*Sih, Bell & Johnson, 2004*; *Réale et al., 2007*), and informs applications such as management of biological invasions (*Sih et al., 2010*). Owing to logistical difficulties inherent to tracking and observing organisms without interference in the field, however, many behavioral studies are conducted *ex situ* in a laboratory setting, where it may be difficult to extrapolate findings to natural conditions (*Niemelä & Dingemanse, 2014*; *Zavorka et al., 2015*). For example, a suite of often-correlated behaviors including aggression, dominance, and boldness are believed to contribute to the success of some invasive over native species (*Pintor, Sih & Kerby, 2009*; *Hudina, Hock & Žganec, 2014*), but these same behaviors can be considerably muted in duration or intensity when observed in the field (*Bergman & Moore, 2003*;

*Larson & Magoulick, 2009*). One of ecology's most fundamental challenges is scaling up from the type of small area and short duration experiments that are easy to conduct, to the larger areas and longer durations over which ecological processes often operate (*Lodge et al., 1998*). This same challenge applies when relating animal behaviors observed in the laboratory to ecological function and intra- or inter-specific interactions *in situ*.

We propose here that linking laboratory behavioral observations to past field interactions or function of specific, individual organisms may be an overlooked application of stable isotope analysis. Stable isotopes of elements such as carbon and nitrogen are assimilated into tissues of consumer organisms relative to their diets in predictable and quantifiable ways (*DeNiro & Epstein, 1978*; *DeNiro & Epstein, 1981*). Importantly, stable isotopes of consumers equilibrate with those of their diets at different rates in different tissues, giving snapshots of ecological interactions that may scale from previous days to years (*Buchheister & Latour, 2010*). Analyzing stable isotope ratios in organisms can provide ecological insights ranging from habitat use and movement (*Hobson, 1999*) to trophic position (*Post, 2002*). For example, stable isotope analysis of feathers has been used to make inferences about migration and habitat use of several species of seabirds that spend winter months far from land and are therefore difficult to study during this period (*Phillips et al., 2009*). In another example, *Cherel et al. (2008)* used stable isotope analysis to identify the trophic position and diet composition of southern elephant seals (*Mirounga leonina*) which forage at depths exceeding 1,000 m and have largely digested their meals by the time they return to land, precluding them from being studied using traditional methods (e.g., direct observation, gut content analysis). Similarly to how these and other studies have applied stable isotope analysis to infer the influence of past behavior on current success of organisms, we propose that stable isotope analysis could permit researchers to link laboratory interactions with previous *in situ* habitat selection, movement, diet choice, or competitive interactions (Fig. 1).

We conducted laboratory behavioral assays to measure individual dominance of invasive rusty crayfish, *Orconectes rusticus*, and used stable isotope analysis to hindcast trophic position of these crayfish under natural field conditions. We predicted that more dominant crayfish in the behavioral assays would have higher trophic positions if dominance were related to competitive ability in the field (e.g., ability to access high quality food such as macroinvertebrates; *Roth, Hein & Vander Zanden, 2006*) or willingness to pursue high-risk, high-reward prey such as fish or other crayfish (*Taylor & Soucek, 2010*). Alternatively, dominance and trophic position may not be associated if laboratory behaviors are ultimately uninformative with respect to past interactions of organisms. Numerous previous studies have used stable isotope analysis to infer various *in situ* behaviors of organisms, such as habitat use and diet preferences (e.g., *Hildebrand et al., 1996*; *Rubenstein & Hobson, 2004*); however, our study is the first to our knowledge to seek a direct relationship for individual organisms between laboratory behaviors and field function as determined by stable isotope analysis, and proposes the linkage of laboratory behavioral assays and stable isotopes as a more common practice in the future.

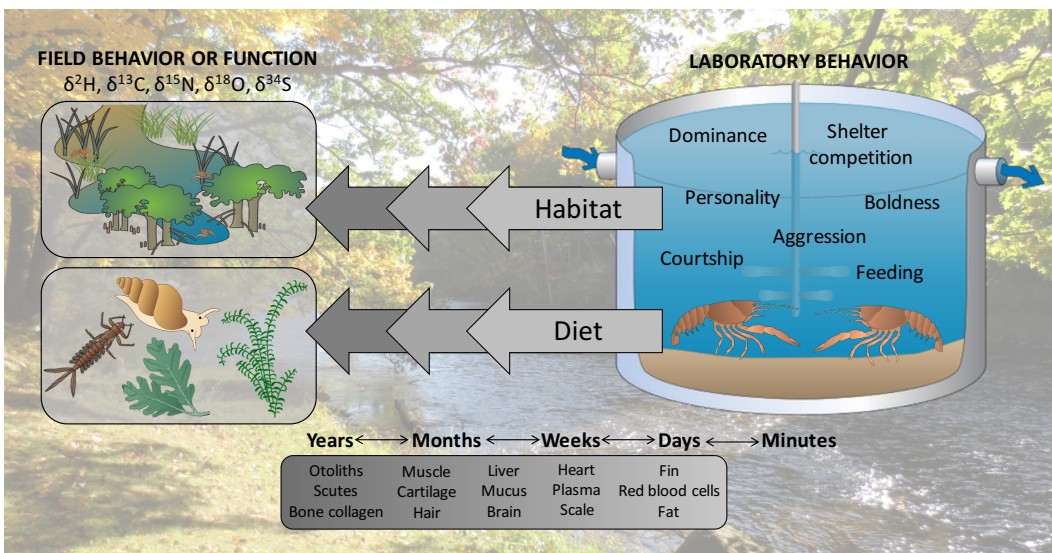

**Figure 1** **Stable isotopes could permit researchers to hindcast the ecological interactions of organisms, linking behaviors observed in the laboratory with previous field function or behavior.** Symbols courtesy of the Integration and Application Network (ian.umces.edu/symbols/). Background image is of the study location where organisms were collected (Chippewa River).

# METHODS

*Orconectes rusticus* was introduced via the bait trade to the Laurentian Great Lakes circa 1960 and has negatively affected fish, macrophytes, and freshwater macroinvertebrates (*McCarthy et al., 2006*; *Peters et al., 2014*). The invasion success of this crayfish has made it the focal point of a large number of laboratory and field studies (e.g., *Olsen et al., 1991*; *Wilson et al., 2004*) and hence, a useful organism to test for linkages between field and laboratory behavior. We collected adult form II (reproductively inactive) male *O. rusticus* ($n = 40$) by hand on 16 June 2015 in the Chippewa River, Michigan (43.5652°, −84.9183°), where this species is invasive. Because size influences the outcome of crayfish agonistic trials (*Bergman & Moore, 2003*), we used crayfish within a carapace length range of 23.41 to 27.53 mm, the smallest size range for which we could collect 40 crayfish (see Supplemental Information for additional morphometrics). *Orconectes rusticus* in this size range are small adults of the same age class (*Momot, 1967*) and are therefore unlikely to have diets that differ from one another due to ontogenetic shifts (*Bondar et al., 2005*; *Larson, Olden & Usio, 2010*). Immediately following collection, we placed crayfish in individual 16 oz. plastic containers filled to a depth of 2 cm with river water and a rock for shelter.

## Agonistic assays

Laboratory agonistic assays for crayfish are often conducted after isolating individuals for at least one week to remove possible previous social experience that could influence interactions (*Seebacher & Wilson, 2007*). We conducted our experiment directly following collection (17 June 2015 during daylight hours [07h19-18h59]), but believe that retaining any existing dominance hierarchies from the field would increase the likelihood of a relationship between laboratory behaviors and previous field function.

**Table 1  Ethogram used to score agonistic assays.** Modified from *Bruski & Dunham (1987)*.

| Score | Description |
| --- | --- |
| −2 | Tail flip or fast retreat |
| −1 | Slow retreat |
| 0 | Within one body length with no visible interaction |
| 1 | Approach without threat display |
| 2 | Approach with threat display (e.g., meral spread, antennal whips) |
| 3 | Boxing, pushing, or other agonistic interaction with closed chelae |
| 4 | Grabbing, tearing, or other agonistic interaction with opened chelae |
| 5 | Full out, unrestrained fighting, usually with interlocked chelae |

We conducted three rounds of twenty, randomized paired assays, with each crayfish fighting one opponent per round (no interactions between individuals were repeated). In order to track individual crayfish, we randomly assigned each crayfish a number from 1 to 40, which we wrote on the dorsal side of its carapace using a permanent marker. Prior to the start of each assay, crayfish were placed on opposite sides of a separator in a 19 l bucket and allowed to acclimate for 15 min. We then removed the separator and allowed the crayfish to interact for 10 min. During each assay, we scored each of the two crayfish individually based on the interactions that took place when they were within one body length of each other. All agonistic assays were watched and scored in real time by a single observer to ensure consistency in scoring. The agonistic assays within each of the three rounds were held in a random order, and the observer had no knowledge of totaled crayfish scores from previous rounds so as to avoid bias.

The scoring system we used has possible point values ranging from −2 (fast retreat) to 5 (unrestrained fighting) and is based on the ethogram modified from *Bruski & Dunham (1987)*; Table 1. Following each assay, the participating crayfish were returned to their original holding container. We then rinsed buckets and refilled them to a depth of 5 cm with fresh water from the Chippewa River (18–20 °C). At the conclusion of all assays, crayfish were placed in individual, labelled bags and euthanized by freezing at −17.8 °C. We calculated the dominance score of each crayfish by first summing its scores from each round, then taking the mean of the three resulting scores.

## Stable isotope analysis

Stable isotope analysis is a technique based on the principle that the ratios of heavy to light isotopes in the tissues of consumers reflect those of their diets in a predictable way (*DeNiro & Epstein, 1978*; *DeNiro & Epstein, 1981*). Stable isotope analysis generally entails drying and homogenizing tissue or whole-body samples of focal organisms, then using a mass spectrometer coupled with an elemental analyzer to determine their constituent ratios of heavy to light isotopes (i.e., $R_{sample}$). The isotope signatures of samples ($\delta^x$), expressed in per mille (‰), are then calculated as $\delta^x = ((\frac{R_{sample}}{R_{standard}}) - 1) \times 1{,}000$ where $R_{standard}$ is the isotopic ratio of a standard (e.g., Vienna PeeDee Belemnite for carbon; air for nitrogen). This technique is often used to study the roles and interactions of organisms in ecosystems,

particularly as related to trophic position and diet composition (*Vander Zanden & Rasmussen, 1999*; *Post, 2002*), but patterns of stable isotope spatial structure can also be applied to study organismal movement and habitat use (*Hobson, 1999*; *Seminoff et al., 2012*).

In freshwater ecology, the most commonly used stable isotopes have been carbon and nitrogen (denoted $\delta^{13}C$ and $\delta^{15}N$, respectively). Specifically, $\delta^{13}C$ provides a tracer of energy source origin because it is fixed by primary producers at photosynthesis and is well-conserved up food chains with little change in value with each increasing trophic level (termed discrimination; generally 0–1‰; *Fry & Sherr, 1984*). Common sources of primary productivity in freshwater habitats that can often be distinguished by analyzing $\delta^{13}C$ include a benthic algal pathway, an open water phytoplankton pathway, and an allochthonous terrestrial detrital pathway; the importance of these pathways to consumers can vary depending on habitat attributes (*Dekar, Magoulick & Huxel, 2009*; *Francis et al., 2011*). In contrast to $\delta^{13}C$, $\delta^{15}N$ can be used to estimate trophic position of organisms as it generally increases or discriminates at a predictable $3.4 \pm 1.1$‰ with each increasing trophic level, from primary producers to primary, secondary, and tertiary consumers (*Minagawa & Wada, 1984*). In some cases, $\delta^{15}N$ can be used alone to infer trophic position of organisms; however, this is not the case if different sources of primary productivity used by a consumer are depleted or enriched in $\delta^{15}N$ relative to each other (*Vander Zanden & Rasmussen, 1999*; *Post, 2002*; Fig. 2). Under these circumstances, mixing models can be used to estimate contributions of different energy pathways to consumers, and subsequently correct for differences in their $\delta^{15}N$ enrichment while calculating trophic position of consumers (*Post, 2002*).

For this experiment, we collected snails (*Elimia livescens*; $n = 45$) and mussels (*Elliptio dilatata*; $n = 5$) in the same stretch of the Chippewa River and on the same date as our crayfish (see above), which we froze at $-17.8$ °C, to be used as primary consumer endpoints in a two end-member stable isotope mixing model related to calculating trophic position of crayfish. We chose these specific organisms as they are reliable primary consumers (i.e., trophic position $= 2$) whose relatively large size and long lives make their isotopic signatures more robust to spatial and temporal variation than those of primary producers (*Cabana & Rasmussen, 1996*; *Post, 2002*). Specifically, we used snails to represent the isotopic signature of algal-based primary production and filter-feeding mussels as an additional endpoint to represent a broad range of other potential sources of primary production in lotic systems (e.g., benthic algae, terrestrial detritus, and phytoplankton from upstream lentic systems; *Raikow & Hamilton, 2001*; *Cole & Solomon, 2012*).

We dissected crayfish for abdominal muscle, snails for whole body without shell, and mussels for foot muscle. We dried samples at 60 °C for 24 h, homogenized them in an ethanol-rinsed mortar and pestle, then weighed and encapsulated aliquots weighing 0.64 $\pm.04$ mg of each sample into tin capsules. We sent these samples to the Stable Isotope Mass Spectrometry Lab at the University of Florida for analysis on a Micromass Prism II isotope ratio mass spectrometer coupled with an elemental analyzer. Two internationally recognized standards (l-glutamic acids), USGS40 (mean $\pm$ standard deviation $\delta^{13}C$, $-26.39$‰ $\pm 0.11$; $\delta^{15}N$, $-4.53$‰ $\pm 0.12$; measured repeatedly for calibration) and USGS41 ($\delta^{13}C$, 47.57‰; $\delta^{15}N$, 37.36‰; measured once as a check standard), were measured during the analysis to ensure precision.

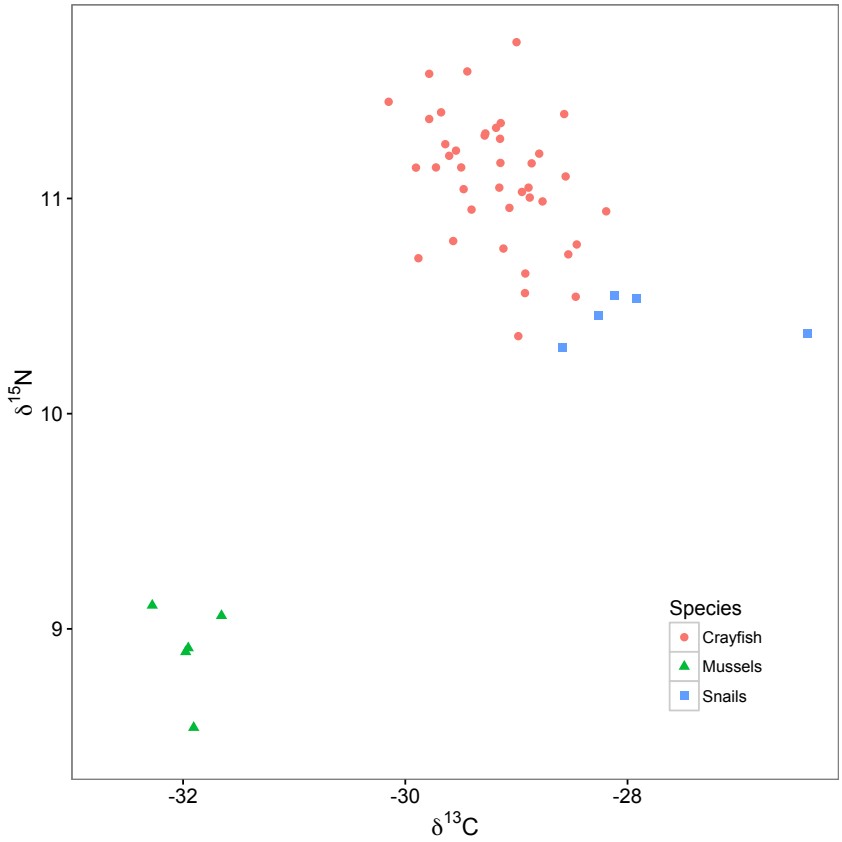

**Figure 2** **Isotopic biplot of $\delta^{13}$C and $\delta^{15}$N for crayfish (red circles), mussels (green triangles), and snails (blue squares).** All values are expressed in per mille (‰) relative to a standard of V-PDB (Vienna PeeDee Belemnite) for carbon and air for nitrogen.

We calculated the relative contribution of the primary productivity represented by snails (SPP) to our crayfish as SPP $= \frac{(\delta^{13}C_{crayfish} - \delta^{13}C_{mussel})}{(\delta^{13}C_{snail} - \delta^{13}C_{mussel})} * 100$, where $\delta^{13}C_{crayfish}$ is the $\delta^{13}$C of each crayfish, $\delta^{13}C_{mussel}$ is the mean $\delta^{13}$C of our mussel samples and $\delta^{13}C_{snail}$ is the mean $\delta^{13}$C of our snail samples. We then calculated the relative contribution of the primary productivity represented by mussels (MPP) as MPP $= 100 - $ SPP. Lastly, we calculated the trophic position (TP) of our crayfish as TP $= 2 + \frac{\delta^{15}N_{crayfish} - (\delta^{15}N_{snail} * \frac{SPP}{100} + \delta^{15}N_{mussel}(\frac{MPP}{100}))}{\Delta^{15}N}$, where $\delta^{15}N_{crayfish}$ is the $\delta^{15}$N of each crayfish, $\delta^{15}N_{snail}$ is the mean $\delta^{15}$N of the snails, $\delta^{15}N_{mussel}$ is the mean $\delta^{15}$N of the mussels, and $\Delta^{15}$N is a trophic discrimination factor of 3.4 (*Minagawa & Wada, 1984*).

## Statistical analysis

We used linear regression to test for a relationship between the mean dominance scores and calculated trophic positions of our crayfish. Additionally, we performed several linear regressions controlling for the effect of body size on crayfish dominance by using residuals, as well as a linear regression testing for a relationship between mean dominance score and unaltered $\delta^{15}$N signatures rather than calculated trophic position (Supplemental Information). All analyses were conducted using the R statistical program (*R Core Team, 2014*).

## RESULTS

Snails were enriched in $\delta^{13}C$ (mean $\pm$ standard deviation; $-27.9 \pm 0.9\text{‰}$; Fig. 2) relative to mussels ($-32.0 \pm 0.2\text{‰}$). The relatively depleted $\delta^{13}C$ signature of the mussels likely reflects utilization of either phytoplankton or allochthonous terrestrial detritus as food resources, relative to the generally more enriched $\delta^{13}C$ signature of the benthic algal pathway (*Raikow & Hamilton, 2001*; *Cole & Solomon, 2012*). The percent reliance of crayfish (mean $\pm$ standard deviation) on the snail pathway was $67.6 \pm 11.3\%$, relative to $32.4 \pm 11.3\%$ on the mussel pathway, indicating that most of these crayfish relied twice as much on the snail than mussel resource pathway (see Supplemental Information for analysis of the relationship between percent reliance of crayfish on the snail primary production pathway and dominance scores). Our mixing model calculations identified reliance of our crayfish on food resources from these two isotopically distinct pathways, allowing us to correct for the relatively depleted $\delta^{15}N$ signature of mussels with respect to the crayfish trophic position. Trophic positions of crayfish ranged from 2.1 to 2.6 with a mean of $2.3 \pm 0.1$, suggesting a range of foraging behaviors from high reliance on primary producers like benthic algae (i.e., trophic position $= 2$) to some predation on primary consumers like snails (i.e., trophic position $= 3$).

The mean crayfish dominance score from the agonistic assays was 29.93 (SD, 28.61; min, $-23.33$; max, 80.67). We did not find a significant relationship between dominance and trophic position ($y = 0.0005x + 2.32$, $R^2 = 0.013$, $F_{1,38} = 0.51$, $p = 0.48$; Fig. 3). Our additional analyses accounting for the role of body size on both dominance and trophic position, as well as those using an alternative measure of trophic position, did not affect our conclusion that there is no association between dominance and trophic position (Supplemental Information).

## DISCUSSION

We failed to find a relationship between crayfish dominance and trophic position. We therefore infer that laboratory dominance among these organisms may not necessarily relate to their dietary preferences in the field, despite our prediction that more dominant crayfish should be more likely than subordinate crayfish to compete successfully for high quality food or to pursue high-risk, high-reward prey (*Roth, Hein & Vander Zanden, 2006*; *Taylor & Soucek, 2010*). However, this is to our knowledge the first attempt to relate laboratory behavior to field performance via stable isotope analysis; therefore, more studies are warranted to further explore linkages between these two techniques in light of possible sources of discord.

For example, other behaviors may correlate better with trophic position than dominance in paired agonistic assays. Dominance assays may instead be more informative with respect to acquisition of shelter to avoid fish predation or fitness via sexual selection (*Garvey, Stein & Thomas, 1994*; *Bergman & Moore, 2003*), whereas trophic position in the field might correlate better with other measures of laboratory behavior, such as boldness. However, dominance and boldness have been observed to correlate as "behavioral syndromes" in crayfish (*Pintor, Sih & Kerby, 2009*), and we would therefore expect boldness and

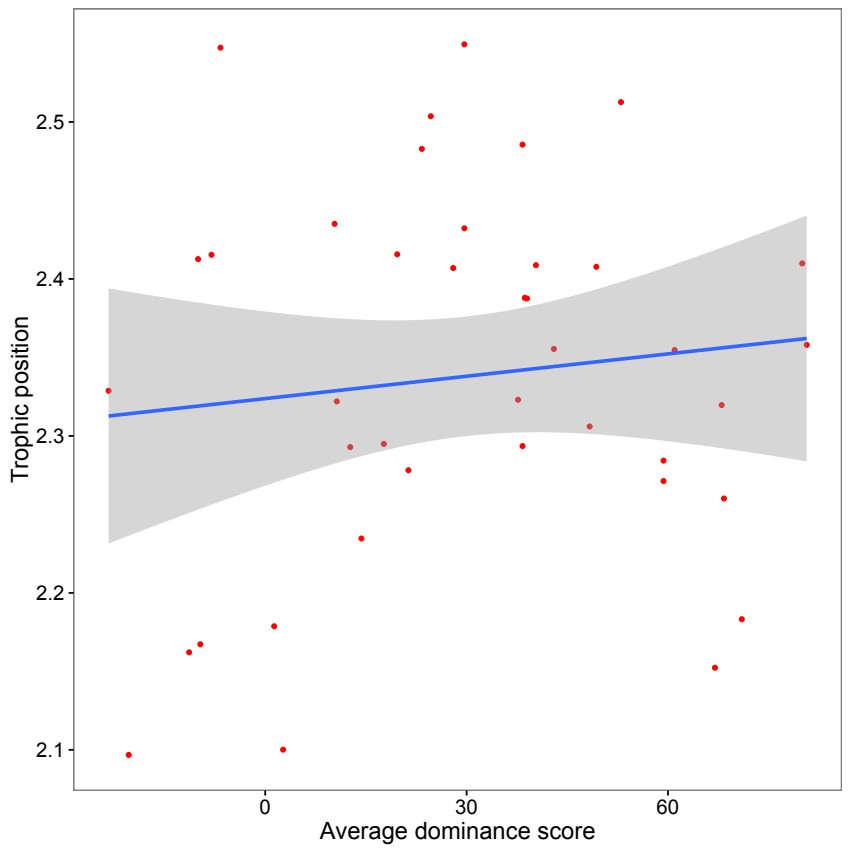

**Figure 3** **Relationship between crayfish trophic position and dominance.** Scatterplot (with 95% CI) of mean assay dominance score for each crayfish over three agonistic assays and *in situ* trophic position ($y = 0.0005x + 2.32$, $R^2 = 0.013$, $F_{1,38} = 0.51$, $p = 0.48$).

dominance to both correlate with trophic position. It is also possible that there is a temporal disconnect between our analysis of crayfish abdominal tissue, which has an isotopic half-life of approximately 20–30 days (*Glon, Larson & Pangle, 2016*), and the social memory of our crayfish, which is thought to last from 60 min to one week (*Bergman et al., 2003*). Use of a tissue with a faster turnover rate (e.g., haemolymph) may better reflect the most recent *in situ* behavior of crayfish. Further, male crayfish of the family Cambaridae cycle between a reproductively inactive form II and an active form I stage. We used form II male crayfish, which are typically less aggressive than crayfish in form I (*Bergman et al., 2003*). Replicating our experiment with form I individuals might alter the results of agonistic assays and their relationship to trophic position.

Lastly, a possible limitation of our study was our relatively small sample size ($n = 40$; *Galván, Sweeting & Reid, 2010*). In order to determine if our lack of a significant relationship stemmed from low power, we conducted power analyses using the pwr package in R (*Champely, 2015*). We found that for our observed effect size (0.013; calculated as $\frac{R^2}{1-R^2}$ *Cohen, 1988*) and an alpha of 0.05 and conventional power of 0.80, we would have required 605 crayfish replicates to observe statistical significance. Conversely, for an alpha of 0.05 and power of 0.80, the smallest effect size we would have detected as significant with 40

crayfish replicates was 0.21 ($R^2 = 0.173$). We therefore conclude that although the effect size observed here could only be detected as statistically significant with an uncommonly high level of replication (perhaps dismissed as statistical significance without biological significance; *Nakagawa & Cuthill, 2007*), our level of replication was adequate to find significant relatively weak effect sizes down to an $R^2 = 0.173$.

Although our study failed to find an association between crayfish dominance and stable isotope-estimated trophic position, we believe that there are many unexplored and promising avenues to combine behavioral and isotope ecology in order to learn more about how behavior observed in laboratories corresponds with movement and organismal interactions in the field. Laboratory experiments and stable isotope analyses have both separately been used to explore the "ecology of individuals" or variation within populations and species (*Bolnick et al., 2003*; *Niemelä & Dingemanse, 2014*; *Zavorka et al., 2015*), yet to our knowledge, researchers have not previously combined or compared these approaches for the same organisms. For example, stable isotope analysis and behavioral assays could be combined to together evaluate whether range expansion of invasive species is being driven by subordinate individuals with low trophic positions that are excluded from core habitats by dominant intraspecific competitors, or instead bold or aggressive individuals with high trophic positions that are inclined to disperse (*Hudina, Hock & Žganec, 2014*). Further, where distinct stable isotope signatures exist over habitat gradients (*Hobson, 1999*), researchers could infer whether individuals with or without dispersal-related behaviors observed in the laboratory were actually recent arrivals or longstanding residents of their collection locations. We encourage future studies to further explore the possible insights gained by linking laboratory behavior with field function through stable isotope analysis, as doing so could contribute meaningfully to an array of ecological and evolutionary questions.

## ACKNOWLEDGEMENTS

We thank Matt Cooper, Jason Curtis, and Jim Student for use of equipment and facilities; Heather Dame and Sara Thoma for help with crayfish collection; and Daelyn Woolnough for assistance with mollusk identification. We also thank Axios Review for their assistance in preparing this manuscript for publication, as well as three reviewers that provided us with helpful feedback.

### Funding

Stable isotope analysis was funded by Central Michigan University. The funders had no role in study design, data collection and analysis, decision to publish, or preparation of the manuscript.

### Grant Disclosures

The following grant information was disclosed by the authors:
Central Michigan University.

## Competing Interests

The authors declare there are no competing interests.

## Author Contributions

- Mael G. Glon conceived and designed the experiments, performed the experiments, analyzed the data, contributed reagents/materials/analysis tools, wrote the paper, prepared figures and/or tables, reviewed drafts of the paper.
- Eric R. Larson conceived and designed the experiments, analyzed the data, wrote the paper, reviewed drafts of the paper.
- Kevin L. Pangle conceived and designed the experiments, analyzed the data, contributed reagents/materials/analysis tools, wrote the paper, reviewed drafts of the paper.

## Ethics

The following information was supplied relating to ethical approvals (i.e., approving body and any reference numbers):

Spike mussels were lawfully collected under a permit from the Michigan Department of Natural Resources.

## Data Availability

The raw data has been supplied as Data S1.

## Supplemental Information

Supplemental information for this article can be found online at http://dx.doi.org/10.7717/peerj.1918#supplemental-information.

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
