# Peer review of "Connecting laboratory behavior to field function through stable isotope analysis"

_PeerJ, doi:10.7717/peerj.1918_

## Round 0.1 · original submission · Minor Revisions

In addition to the reviews you supplied from Axios, I asked for an additional review from an expert on stable isotopes (Reviewer #1). That reviewer found your manuscript to be of interest and scientifically sound. There are, however, some modest issues that need your attention before acceptance. Please see the reviewer's specific comments and address those in a quick revision. That reviewer also suggests one additional contrast to make using your data. I find that suggestion reasonable, but will allow you to include that contrast, or not, at your discretion.

·

Basic reporting

This is a clearly-written paper that presents an interesting test of relationships between isotopically measured trophic position and behavior in crayfish.

Experimental design

I'm not an expert in behavioral studies of this sort, but the behavioral assays seem reasonable to me. The isotopic measurements and mixing models are appropriate for the questions asked.

Validity of the findings

The findings, as presented, appear robust. I do wonder why the authors didn't look at the potential relationship between diet type (not just TL) and agonistic behavior. They have the data and may have already done so, but it seems to me that animals that differ in aggression might exploit different types of food resource.

Additional comments

This paper describes an interesting mix of behavioral and isotopic work and Glon et al. are correct that this appears to be a novel approach of potentially broad application. Although their primary result was a negative one (no association between agonistic behavior and trophic position), this paper should be of interest to both behavioral ecologists and isotope ecologists with an interest in freshwater systems. I suggest looking at the interaction between diet type (snail vs mussel food webs) and agonistic behavior, or at least mentioning the results of that comparison if it's already been done.

I do have a number of specific comments suggestions for the authors, which are detailed below.

Specific Comments

Materials and Methods
Lines 188-189: this system will provide d13C values along with the d15N measurements. Why not construct a mixing model for 13C as another index to food sources?

Line 198: The eqn for TP is incorrect as written – SPP values have to be divided by 100 for this to work.

Results
Lines 215-216: I don't understand the sentence that begins "Our mixing model…"
The mixing model simply allows you to estimate the relative contribution to crayfish tissues of the two isotopically distinct end members. Or did I miss something here?

Lines 227-228: What about dietary type? Is there any association between dominance and the types of food (not TL, but rather food derived from the benthic vs planktonic sources) consumed?

Discussion
Lines 267-268: Did TL scale with animal size? This is a useful test since larger animals should have access to a broader range of potential pretty. OK, I now see that you address this in lines 578-579, but this is an important point that should be mentioned in the main body of the paper.

Figures and Tables
Figure 2: This crossplot shows very nice separation of the three species, but the axis labels are too small for easy legibility.

Figure 3: Here again, the axis labels are far too small.

---

## Round 0.2 · accepted · Accept

Thanks for your revision. The MS is now acceptable for publication.

Congratulations

External reviews were received for this submission. These reviews were used by the Editor when they made their decision, and can be downloaded below.